# *ENPP2* Promoter Methylation Correlates with Decreased Gene Expression in Breast Cancer: Implementation as a Liquid Biopsy Biomarker

**DOI:** 10.3390/ijms23073717

**Published:** 2022-03-28

**Authors:** Maria Panagopoulou, Andrianna Drosouni, Dionysiοs Fanidis, Makrina Karaglani, Ioanna Balgkouranidou, Nikolaos Xenidis, Vassilis Aidinis, Ekaterini Chatzaki

**Affiliations:** 1Laboratory of Pharmacology, Medical School, Democritus University of Thrace, 68100 Alexandroupolis, Greece; mpanagop@med.duth.gr (M.P.); androsouni@gmail.com (A.D.); mkaragla@med.duth.gr (M.K.); ioannabio@yahoo.it (I.B.); 2Institute of Agri-Food and Life Sciences, Hellenic Mediterranean University Research Center, 71410 Heraklion, Greece; 3Institute of BioInnovation, Biomedical Sciences Research Center Alexander Fleming, 16672 Athens, Greece; fanidis@fleming.gr (D.F.); aidinis@fleming.gr (V.A.); 4Department of Medical Oncology, Democritus University of Thrace, 68100 Alexandroupolis, Greece; nxenidis@freemail.gr

**Keywords:** autotaxin, *ENPP2*, methylation, breast cancer, liquid biopsy, expression, regulation

## Abstract

Autotaxin (ATX), encoded by the ctonucleotide pyrophosphatase/phosphodiesterase 2 (*ENPP2*) gene, is a key enzyme in lysophosphatidic acid (LPA) synthesis. We have recently described *ENPP2* methylation profiles in health and multiple malignancies and demonstrated correlation to its aberrant expression. Here we focus on breast cancer (BrCa), analyzing in silico publicly available BrCa methylome datasets, to identify differentially methylated CpGs (DMCs) and correlate them with expression. Numerous DMCs were identified between BrCa and healthy breast tissues in the gene body and promoter-associated regions (PA). PA DMCs were upregulated in BrCa tissues in relation to normal, in metastatic BrCa in relation to primary, and in stage I BrCa in relation to normal, and this was correlated to decreased mRNA expression. The first exon DMC was also investigated in circulating cell free DNA (ccfDNA) isolated by BrCa patients; methylation was increased in BrCa in relation to ccfDNA from healthy individuals, confirming in silico results. It also differed between patient groups and was correlated to the presence of multiple metastatic sites. Our data indicate that promoter methylation of *ENPP2* arrests its transcription in BrCa and introduce first exon methylation as a putative biomarker for diagnosis and monitoring which can be assessed in liquid biopsy.

## 1. Introduction

Breast cancer (BrCa) is one of the most common cancers in the world among women [1]. Currently, early detection and new treatment options have improved the survival rate; however, clinical challenges still persist due to drug resistance and relapse being the leading cause of morbidity and mortality [2,3,4]. There is still an emerging need to define the biological mechanisms associated with the pathogenesis of BrCa and identify biomarkers and targets to improve treatment strategies.

The ATX-LPA signaling axis attracts growing interest in cancer research [5]. ATX is a secreted catalytically active glycoprotein that belongs to the ectonucleotide pyrophosphatase/phosphodiesterase (ENPP) family and is encoded by the *ENPP2* gene [5,6]. ATX has a lysophospholipase D activity and is mainly responsible for catalyzing the hydrolysis of extracellular LPC into LPA [5,6]. LPA then acts through at least six G-coupled receptors (GPCRs), known as LPAR1-6, and can activate various signaling pathways in almost every mammalian cell type [7]. Breast carcinogenesis was first linked to ATX and LPA signaling back in 1995, by observations that ATX promotes proliferation of breast and ovarian cancer cells [8]. Since then, several studies have associated aberrant expression of ATX and LPA signaling with BrCa pathogenesis and metastatic progression [5,9,10,11,12].

Recently, a few studies reported that the ATX-LPA axis is governed by epigenetic regulation of the gene encoding ATX, ectonucleotide pyrophosphatase/phosphodiesterase 2 *(ENPP2)*. DNA methylation is a well-studied epigenetic mechanism that regulates expression [13]. The identification of abnormal methylation in tissue or liquid biopsy has been correlated to cancer initiation and progression [14,15,16,17]. By employing an in silico approach, we have recently described *ENPP2* methylation profiles in health and malignancy, showing that methylation is an active level of ATX expression regulation in cancer. Increased methylation of promoter and first exon cytosine-guanine dinucleotides(CGs) and respective decreased *ENPP2* mRNA expression were found in prostate and lung cancers and were correlated to poor prognostic parameters [18].

Here, we focus in BrCa, presenting an in silico methylation and expression analysis of *ENPP2* followed by an experimental investigation in liquid biopsy. In specific, we explored the methylation status of *ENPP2* in BrCa and correlated it to its expression by using publicly available high-throughput methylation datasets from the Illumina methylation 450 bead-chip array, found in The Cancer Genome Atlas (TCGA) and the Gene Expression Omnibus (GEO) databases. Retrieved data were allocated into groups according to four important clinical endpoints related to prognosis and diagnosis to conduct differential methylation and expression analysis. *ENPP2* methylation, expression at protein and mRNA levels and survival were also estimated using the UALCAN platform. Identified differentially methylated *ENPP2* CGs were further validated in patient ccfDNAs to evaluate their potential for clinical implementation in liquid biopsies for the diagnosis and prognosis in BrCa.

## 2. Results

### 2.1. In Silico Analysis of ENPP2 Methylation and Expression in BrCa

#### 2.1.1. Differential Methylation and Expression Analysis between BrCa and Normal Breast Tissue

Raw methylome data from 520 BrCa (primary and metastatic) and 185 normal breast tissues were analyzed for the 14 CGs of *ENPP2* that the Infinium Human Methylation 450 k platform contains by means of RnBeads. In total, ten DMCS (FDR < 5 × 10^−2^) were detected among breast tissues from healthy individuals and BrCa patients (Table 1). In promoter-associated (PA) regions known to be strongly associated with regulation of expression by methylation, i.e., transcription start site (TSS) and first Exon [19,20], all DMCs (cg04452959, cg02709432, cg02156680, cg06998282, and cg02534163) presented increased methylation in BrCa in relation to normal breast tissue. In the gene body, two CGs (cg00320790, cg20048037) were hypomethylated in BrCa and three CGs (cg09444531, cg26078665, cg23725583) were hypermethylated.

In order to address if the observed aberrant methylation of *ENPP2* in BrCa is associated with alterations in gene expression, we examined *ENPP2* mRNA levels in the same TCGA samples (GEO samples excluded as no expression data were available). Comparisons were made between 302 BrCa and 76 normal breast tissues. Results showed downregulation of expression in BrCa in relation to normal tissues (FC:−5.15, FDR:3.96 × 10^−66^) (Table 2), indicating that the increased methylation of *ENPP2* in promoter and first exon regions is correlated with lower gene expression in BrCa. Discrete samples distribution based on methylation and expression is depicted in Figure 1. In specific, Figure 1A shows the distribution of BrCa and healthy samples in reduced dimensional space. Pathology is the main source of variation in both expression and methylation values. Correlation of *ENPP2* mRNA expression per CG methylation revealed statistically significant correlations, showed in Figure 2A and Table 3. In BrCa tissues, important correlations emerged only for PA CGs, as for TSS CGs (cg06998282, cg14409958) and first exon CG (cg02534163), methylation showed a reverse correlation with expression. These results further confirm that increased methylation at these gene regions is associated with decreased expression. Analysis between *ENPP2* methylation and expression in normal breast tissues showed a negative correlation for two gene body CGs, namely cg07236691 and cg2372583, and a positive correlation for the gene body cg09444531 (Table 3).

#### 2.1.2. Differential Methylation Analysis between Primary and Metastatic BrCa

Methylomes of primary BrCa were analyzed in comparison to those from metastatic BrCa in order to detect changes in *ENPP2* related to metastatic transformation. Raw data from 132 primary cancers and 31 cancers with distant metastasis were analyzed using RnBeads and 6 DMCs out of a total of 14 CGs (FDR < 5 × 10^−2^) were detected (Table 1). Four of them were located at the gene body (cg20048037, cg09444531, cg26078665, cg23725583) presenting lower methylation and two were located in the TSS and first exon (cg06998282 and cg02534163, respectively) showing upregulation in metastatic in relation to primary BrCa. These observations suggest an involvement of *ENPP2* methylation in BrCa progression and metastasis.

#### 2.1.3. Differential Methylation and Expression Analysis between Stage I BrCa and Normal

In order to address if aberrant *ENPP2* methylation is an early effect in the breast carcinogenetic process, methylome raw data from 136 stage-I BrCa and 111 normal breast tissues were subjected to RnBeads differential methylation analysis. A total of 9 out of the 14 studied CGs were DMCs between stage I BrCa and normal tissues (Table 1). All but one gene body DMCs and all five PA DMCs showed increased methylation in stage I BrCa in relation to normal tissues.

Differential mRNA expression analysis between 111 Stage I BrCa and 66 Normal breast tissue samples showed downregulation of expression in Stage I cancer (FC:−5.46, FDR:3.43 × 10^−52^) (Table 2), similarly with findings of the analysis between all BrCa samples and normal tissues. Dimensionality reduction plot (Figure 1B) depicts excellent separation of Stage I BrCa and normal samples based on level 3 methylation and normalized expression values. Interestingly, a negative correlation between methylation and expression was noted for all TSS (cg02156680, cg04452959, cg06998282, cg14409958) and first exon CGs (cg02534163) in BrCa samples and no correlation emerged for gene body CGs (Table 3). A different pattern was observed for control samples as methylation of three gene body CGs, namely cg07236691 (negatively), cg23725583 (negatively), and cg09444531 (positively), were found to be correlated to mRNA expression. Finally, one TSS CG (cg14409958) presented reverse correlation between methylation and expression (Table 3).

#### 2.1.4. Differential Methylation and Expression Analysis between Early- and Advanced-Stage BrCa

In order to detect important methylation events related to the progression of BrCa to advanced-stage disease, we conducted an analysis of raw methylome data from 521 early (stage I, II) and 221 advanced (stage III) BrCa patients. Only two DMCs (cg01243251, cg20162626) were identified in the gene body region, showing a slight but still statistically significant decrease in methylation in advanced in relation to early BrCa. No difference was observed in TSS or first exon CGs (Table 1). In accordance to methylation, no difference in mRNA expression was observed between 519 stage I and II cancers and 191 stage III cancers (FC: 1.20, FDR: 9.41 × 10^−2^) (Table 2). Last, in dimensionality reduction plots (Figure 1C) there was no separation of early and advanced stage BrCa samples based on detected features’ methylation and expression.

As expected, correlation analysis revealed important relationships for PA CGs. In specific, in advanced cancer, methylation of three TSS (cg04452959, cg06998282, cg14409958) and one first exon CG (cg02534163) was negatively correlated with *ENPP2* expression (Figure 2C and Table 3). Same tendency was observed in the early cancers except for the case of cg04452959 (Figure 2C and Table 3).

Cumulatively, previous analysis indicated that *ENPP2* methylation is associated with the malignant transformation of breast cells and metastasis. However, minimal methylation changes were noted between stage I/II and stage III BrCa. Similarly, downregulation of *ENPP2* expression was noted in BrCa samples in relation to control but not between early and advanced stage.

#### 2.1.5. Differential Methylation Analysis between BrCa Cancer Types

We also examined differences in the methylation of *ENPP2* between BrCa cancer types. In particular, differential methylation analysis was performed between 473 invasive ductal and 186 invasive lobular BrCa (accounting for 90.2% of available TCGA cases). Three CGs of *ENPP2* were found differentially methylated (Table 4), showing small but statistically significant differences. In specific, body CGs were either hyper-(cg01243251) or hypo-methylated (cg20048037) in ductal in relation to lobular BrCa. A third CG located at the TSS1500 (cg02156680) presented downregulation of methylation in the ductal type.

#### 2.1.6. In Silico Analysis of *ENPP2* Methylation in BrCa ccfDNA Data

The analysis of a ccfDNA dataset (GSE1222126) revealed eight *ENPP2* DMCs (Table 5) between BrCa patient-derived samples and healthy individuals. In specific, four (cg04452959, cg02156680, cg06998282, cg14409958) out of five studied TSS CGs and one first exon CG (cg02534163) were found hypermethylated in BrCa ccfDNAs in relation to control. In the gene body, two (cg07236691, cg20162626) and one (cg20048037) CGs were hypomethylated and hypermethylated, respectively, in BrCa. It has to be noted that four DMCs in PA regions were common between BrCa ccfDNA and breast tissue samples (clinical endpoint: BrCa vs. normal), suggesting that ccfDNA may reflect the methylation status of the tumor.

### 2.2. ENPP2 Methylation, Expression and Survival Analysis by UALCAN

In order to further verify our findings, we conducted ENPP2 expression, methylation, and survival analysis in BrCa using the UALCAN platform. Analysis confirmed above results, as promoter methylation level of *ENPP2* was increased in primary tumor tissues of BrCa patients in relation to normal (*p* < 1 × 10^−12^) as depicted in Figure 3A. Next, expression analysis showed downregulation of ENPP2 mRNA expression in primary tumor tissues in relation to normal tissues (*p* = 1.6 × 10^−12^) as depicted in Figure 3B. Similarly, protein expression analysis showed downregulation in primary tumors in relation to normal tissues (Figure 3E, *p* = 4.5 × 10^−12^). Methylation and expression results by UALCAN strengthen our findings, showing that the *ENPP2* gene is methylated in BrCa and this is related to lower expression, suggesting a causative relationship and a methylation regulatory mechanism in BrCa. Finally, survival analysis did not reveal any statistical significance as depicted in Appendix A.

### 2.3. Methylation Analysis of ENPP2 in ccfDNA from BrCa Patients

Following the in silico analysis, the methylation of *ENPP2* was investigated in BrCa patient-derived ccfDNAs and was compared to their healthy counterparts, using qMSP, in order to test clinical applicability in liquid biopsy. Primers were designed to include the cg02534163 of the first exon of *ENPP2*, a CG identified as a DMC in the in silico analysis in both tissues, and ccfDNA and could be exploited as a biomarker in BrCa.

*ENPP2* methylation was investigated in ccfDNAs isolated from 52 adjuvant, 19 metastatic, and 15 neoadjuvant BrCa patients and 20 healthy individuals (control). Methylation was detected more often in ccfDNA of BrCa patients than in healthy individuals in a statistically significant manner (*p* = 1 × 10^−2^) (Figure 4A). In specific, methylated *ENPP2* was detected in 21 out of 46 (45.6%) of control samples and in 62 out of 86 (72.1 %) of BrCa samples. Between BrCa groups, methylation was detected in 71.1% (36/52), 73.6% (14/19), and 80% (12/15) of adjuvant, metastatic, and neoadjuvant groups, respectively, showing statistically significant differences between groups (*p* = 2 × 10^−2^). In specific, when methylation positives of each group were compared separately, statistically significant correlations emerged between control and adjuvant (*p* = 9 × 10^−3^), control and metastatic (*p* = 3.6 × 10^−2^), control and neoadjuvant (*p* = 2 × 10^−2^), but not between BrCa groups.

Next, methylation levels were also measured. Methylation levels were found elevated in ccfDNA of BrCa patients compared to ccfDNA of healthy individuals, but no statistically significant correlation emerged (*p* = 8 × 10^−2^) (Figure 5A). Between groups, significantly increased levels of *ENPP2* methylation were found in the neoadjuvant group as compared to the control group (*p* = 8 × 10^−4^) and the adjuvant group (*p* = 1 × 10^−3^) (Figure 5B). This result could be due to the fact that neoadjuvant patients have increased tumor burden in relation to adjuvant patients, having their tumor removed. No other statistically significant difference in methylation levels were observed between the other studied groups (Control vs. Adjuvant: 4.3 × 10^−1^, Control vs. Metastatic: 4.5 × 10^−1^, Adjuvant vs. Metastatic: 8.4 × 10^−1^, Metastatic vs. Neoadjuvant: 5.5 × 10^−1^) (Figure 5B). A significant correlation was found in the metastatic group, as patients with one metastatic site presented lower methylation levels than those having more metastatic sites (*p* < 1 × 10^−2^) (Figure 5C). Finally, no other correlation emerged between *ENPP2* methylation and grade, nodal status, tumor size, or age.

## 3. Discussion

ATX is a well-known enzyme responsible for generating lysophosphatidic acid (LPA) and dysregulation of its expression has been linked to several pathologies and to cancer [5,6]. ATX’s encoding gene *ENPP2*, is epigenetically regulated, as aberrant methylation patterns were described in five different cancer types and were correlated to mRNA expression. Furthermore, increased methylation of *ENPP2* was connected to poor prognostic parameters [18]. In most cancer types, ATX is increased, but BrCa cells express little ATX. Still, ATX from its microenvironment plays an important role in BrCa development, promoting cell proliferation, migration, and survival, and is also regarded as a potential target for therapy or increased chemotherapeutic sensitivity [21].

In the present study, we focus on *ENPP2* methylation and expression in BrCa. We first adopted a bioinformatic approach using publicly available datasets of BrCa tissues and ccfDNA. Among 10 DMCs identified between BrCa and normal tissues, all TSS and 1st Exon DMCs presented increased methylation in BrCa. These gene regions are known to be strongly associated with regulation of expression by methylation [19,20] indicating epigenetic arrest of *ENPP2* transcription in BrCa. Indeed, expression analysis showed decreased transcription in BrCa in relation to normal tissues, and increased methylation of all PA CGs was reversely correlated to mRNA expression, results also confirmed by analysis in the UALCAN platform. These results agree with our previous findings in HCC, melanoma, CRC, LC, and PC, showing *ENPP2* hypermethylation in PA and decreased expression [18].

In addition, *ENPP2* is one of 11 genes spanning the 8q12.1-q24.22 genomic region found to be differentially methylated and expressed in invasive breast carcinomas harboring the 8p11-p12 amplicon by integrative analysis. In accordance to our results, *ENPP2* was the only gene showing lower expression levels and hypermethylation despite amplification of the 8p11-p12 amplicon [22]. In fact, one of the DMCs identified here in BrCa and previously in HCC and PC, located in TSS1500 (cg02156680) was included in an eight-feature methylation-based breast-cancer specific signature constructed by integrative analysis of genome-wide DNA methylation and was shown to reliably separate BrCa from normal samples [23]. In contrast, the whole *ENPP2* gene was not included in the methylation-based biosignatures of translational relevance built via automated machine learning analysis of BrCa whole methylome datasets and was not identified as a top rated DMG [24].

When compared to healthy tissues, stage I BrCa showed hypermethylation in PA CGs which were correlated to downregulation of expression, indicating an early event in BrCa. However, when comparison was made between early (I, II) and advanced (III) stages of BrCa, no difference was observed in the methylation pattern of PA CGs, but only in two gene body CGs, and no changes in mRNA expression, suggesting minor *ENPP2* methylation events in the course of the disease. Another study based on TCGA datasets analysis reported *ENPP2* as one of 66 significantly hypermethylated genes with logFC > 1.8 between Stage I–III BrCa [25].

Comparison between primary and metastatic BrCa revealed six *ENPP2* DMCs. Among them, two were in TSS and first exon showing hypermethylation of *ENPP2* in metastatic BrCa, implying a participation in the metastatic cascade. Increased expressions of ATX in the stroma is associated with aggressiveness of human BrCa in women [26], whereas *ENPP2* is one of the 40–50 most up-regulated genes in metastatic solid tumors [5]. Unfortunately, expression data from metastatic samples were not available to allow correlation with methylation and deeper understanding in the process of metastasis. It has to be noted that in our previous work, *ENPP2* hypermethylation in lung cancer was correlated with advanced cancer stage [18].

Small differences in *ENPP2* methylation were observed between ductal and lobular BrCa pathological types, and no further analysis was possible due to lack of relevant clinical information in the available datasets and small representation from other BrCa types. GEO datasets, for example, do not include the cancer type as a parameter for each case. This underlines the significance of the integrity of information provided in the archived datasets to allow full exploitation of readings towards clinical relevance.

*ENPP2* hypermethylation of PA associated CGs was also detected in methylome datasets of ccfDNAs from BrCa patients in relation to ccfDNA from healthy individuals, presenting a similar profile as in the case of tissue samples. Taken together, these findings suggest that assessing methylation in ccfDNA can dynamically reflect methylation events of the tumor. This notion is further supported by our recent in vitro study showing that the methylation profile of ccfDNA released by breast and cervical cancer cell lines is identical to their genomic DNA [27]. Similarly, in CRC, we detected identical methylation profiles of corticotropin releasing factor receptor genes in tumors as in patient ccfDNAs [16].

We further evaluated *ENPP2* methylation in ccfDNA of BrCa patients in order to examine its clinical value as a biomarker. The first exon cg02534163 was targeted in a qMSP assay, chosen because it was identified as a DMC in all clinical endpoints examined, except in the case of early vs. advanced disease, presenting the highest FDR. Analysis showed that *ENPP2* hypermethylation was detected more often in ccfDNA of BrCa patients than in healthy individuals in a statistically significant manner. In a previous study addressing *ENPP2* methylation in ccfDNA from 22 healthy and 45 Taiwanese BrCa patients, no significant differences were found [28], although *ENPP2* methylation showed a twofold increase in BrCa in relation to adjacent normal tissue. Methodological differences or even population genetic variations might explain these different findings. Importantly, in our study, ccfDNA methylation levels of *ENPP2* were also elevated in the neoadjuvant and metastatic groups of patients in relation to adjuvant and control group of patients. This result could be due to the fact that in the new adjuvant and metastatic groups, patients still have a significant tumor burden. Our bioinformatic analysis also showed that *ENPP2* methylation is increased in metastasis in relation to primary cancers. In addition to that, according to our experimental analysis, patients having two or more metastatic foci presented more increased *ENPP2* methylation levels than those patients having a distant metastasis in one organ. Cumulatively, our experimental results are in accordance with those from bioinformatic analysis showing hypermethylation of *ENPP2* in BrCa tissue and ccfDNA and a correlation with cancer aggressiveness and metastasis, suggesting its potential as a novel circulating biomarker in BrCa.

A limitation of our study is the small number of patients enrolled in the experimental part not allowing significant correlations between *ENPP2* methylation levels and clinicopathological features such as grade and tumor size to emerge. Future validation in a larger group of patients should be conducted in order to confirm its clinical value. Furthermore, studies employing in vitro models should clarify the connection between methylation and expression during the carcinogenic process.

## 4. Materials and Methods

### 4.1. Bioinformatic Analysis of ENPP2 in BrCa

#### 4.1.1. Data Sources

Raw DNA methylation data from BrCa tissues and normal breast tissues as well as the corresponding clinical and demographic data were obtained from The Cancer Genome Atlas (TCGA) [29] and Gene Expression Omnibus (GEO) [30] databases. TCGA case inclusion criteria were: 1. Platform: Infinium Human Methylation 450 K bead-chip 2. Primary site: breast; 3. Project: TCGA-BRCA; 4. Gender: female; 5. Age at diagnosis: 26–80 years; 6. Race: white, black or African American, Asian, and not reported. A total of 730 cases were downloaded. As for the BrCa type at diagnosis, the majority of the cases were invasive ductal (64.7% of total cases) or lobular (25.5% of total cases) BrCa, 9.8% of cases were of eight other BrCa types (e.g., secretory, tubular, papillary, and others). The GEO database was searched using ‘Breast cancer’, ‘Metastatic Breast cancer’, ‘cell free DNA’ as keywords and ‘Methylation profiling by array’ as study type. In total, 96 studies were found. Those using the Infinium Human Methylation 450 K bead-chip array and providing adequate raw and clinical data were selected for further analysis, i.e., five studies, namely GSE72245, GSE72251 [31], GSE88883 [32], GSE108576 [33], GSE74214 and GSE122126 [34]. Analysis of *ENPP2* methylation in tissues was performed against 4 major clinically relevant endpoints, as presented in Table 6. Analysis of *ENPP2* methylation was also performed in a ccfDNA dataset (GSE122126) [34] including three BrCa ccfDNA samples and two from healthy individuals.

#### 4.1.2. Data Preprocessing and DNA Methylation Analysis

Raw DNA methylation data (IDAT files) and sample annotation files were subjected to the Bioconductor R package RnBeads v2.0 [35]. RnBeads is a software tool suitable for large-scale analysis, interpretation, and visualization of DNA methylation data. In our workflow, *ENPP2* CGs were chosen as the genomic region of interest and were analyzed for each of the four endpoints, as previously reported by our team [16,24]. Beta methylation values are expressed as decimal values between 0.0 (no methylation) and 1.0 (full methylation). DMCs (DMCs) for *ENPP2* were identified based on the false discovery rate (FDR-adjusted *p*-value < 5.00 × 10^−2^).

#### 4.1.3. Differential Expression Analysis and Expression—Methylation Correlation

Raw RNA-seq (Illumina HiSeq) paired with level 3 methylation data described above (Table 6) were obtained from the TCGA database. No expression data were available for the GEO retrieved methylation datasets. In detail, 66 Normal, 302 BrCa, 111 Stage I, 191 advanced, and 519 early-stage samples were obtained. Expression data were EDASeq normalized and quantile filtered post to differential expression analysis using the edgeR package. Absolute fold change ≥ 1.2 and adjusted *p*-value ≤ 5 × 10^−2^ were selected as thresholds of significant differential expression. Multi-dimensional scaling (MDS) plots of normalized expression and principal component analysis (PCA) of level 3 methylation data were created to visualize sample separation according to the phenotype attributed. Normalized expression and level 3 methylation data were correlated using the Spearman method. Absolute rho value ≥ 0.4 and adjusted *p* value ≤ 5 × 10^−2^ were set as significant correlation thresholds. All the above manipulations were performed with TCGA biolinks package version 2.18.0 [36] and R version 4.0.4.

#### 4.1.4. Expression, Methylation and Survival Analysis Using the UALCAN Platform

In order to further verify our results, we used the UALCAN platform [37] that enable researchers to analyze cancer archived omics data. We performed expression, promoter methylation, and survival analysis of *ENPP2* gene in BrCa and corresponding controls. According to UALCAN, different beta value cut-offs have been considered to indicate hyper-methylation (Beta value: 0.7–0.5) or hypo-methylation (Beta-value: 0.3–0.25). For mRNA expression, methylation, and survival, we used TCGA gene analysis, and the screening conditions were as follows: gene “*ENPP2*”, TCGA dataset “Breast Cancer” and then we used “expression”, “methylation”, and “survival” as links for analysis. Protein expression analysis was performed using the Clinical Proteomic Tumor Analysis Consortium (CPTAC) datasets, Z-values represent standard deviations from the median across samples for the given cancer type. Log2 Spectral count ratio values from CPTAC were first normalized within each sample profile then normalized across samples.

### 4.2. Methylation Analysis of ENPP2 in BrCa Liquid Biopsies

#### 4.2.1. Study Groups and Clinical Samples

The study was approved by the Scientific Board of the University General Hospital of Evros (PGNE), following assessment by Ethics Committee (decision 663/08.08.16), and was conducted according to the ethical principles of the 1964 Declaration of Helsinki and its later amendments. All patients participated after signing a voluntary informed consent.

Blood samples were collected from 86 BrCa patients who visited the Department of Medical Oncology of PGNE and were allocated to three groups: (a) 52 patients having recently (within the previous month) undergone surgery for primary BrCa, exactly before the initiation of adjuvant therapy (adjuvant group), (b) 15 patients upon diagnosis for BrCa, having no previous surgery, before the initiation of neo-adjuvant therapy (neo-adjuvant group), (c) 19 patients upon diagnosis for metastatic disease before the initiation of first-line chemotherapy (a combination of Taxane/Anthracyclines) (metastatic group). Pathological BrCa type was invasive ductal carcinoma for all patients enrolled in the study. The available clinicopathological features for all patient groups are presented in Table 7. Five-year follow-up data were also available. In the adjuvant group, 10 (19.23%) patients have died as a consequence of their disease progression and respective numbers in the metastatic and neo-adjuvant BrCa groups were 11 (57.89%) and 5 (33.33%), respectively.

Peripheral blood was collected in EDTA before treatment and was processed immediately for plasma isolation. In parallel, blood samples from 46 age-matched healthy female donors were included in our study (mean age: 55.65 (±SD) (±13.7), median: 57.0 (range: 26.0–83.0)) (control group). All blood samples were centrifuged within 2 h twice at 2000× *g* and then at 10,000× *g* for 10 min and plasma was stored at −80 °C until further use.

#### 4.2.2. ccfDNA Extraction

ccfDNA from plasma was extracted using the QIAamp DNA Blood Mini kit (Qiagen, MD, USA) according to manufacturer’s instructions with some modifications. Specifically, DNA was eluted from 500 μL of plasma in 25 μL elution buffer and then stored in −20 °C until further use.

#### 4.2.3. Sodium Bisulfite Conversion of ccfDNA

Bisulfite conversion was performed by EZ DNA Methylation-Gold™ Kit (ZYMO Research Co., Orange, CA, USA) as described by the manufacturer. During conversion, all unmethylated but not the methylated-cytosines of ccfDNA were converted to uracil. DNA was then eluted in 10 μL elution buffer and stored at −80 °C until further use. In each experiment, CpGenome Human methylated and nonmethylated DNA standards (Merck Millipore, Darmstad, Germany) or H_2_O were included as positive and negative controls respectively.

#### 4.2.4. Quantitative Methylation-Specific PCR (qMSP)

Methylation of *ENNP2s* 1st Exon was analyzed by qMSP. *ENPP2* primers (Table 8) for methylated sequences specifically designed to contain cg02534163 were checked using Oligo 7 software and obtained from Eurofins, Genomic (Louisville, USA). A methylation-independent assay with non-CpG including primers for the β-actin gene (*ACTB*) was used in order to verify DNA quality and to normalize results. Specificity and cross-reactivity of methylated primers were evaluated by using SB-converted methylated and non-methylated DNA standards. Analytical sensitivity of qMSP assays was evaluated by using serial dilutions of SB-converted methylated and nonmethylated DNA standards (100%, 50%, 10%, 1%, 0%). The assay efficiency (expressed as E = 10^−1^/slope^−1^) was evaluated by using serial dilutions of the SB-converted methylated DNA standards in H_2_O (100–0.01 ng). The results were calculated using the Rotor-Gene 6000 Series Software 1.7 (Qiagen). The analysis was performed according to the RQ sample (Relative Quantification) = 2^−ΔΔCT^ method. Specifically, ΔΔCT values were generated for each target after normalization by *ACTB* values and using 100% methylation as calibrator.

In the *ENPP2* qMSP assay, a high linear correlation was found between the dilution ratios, analytical sensitivity of 0.01%, and efficiency 99%. Curves are presented in Appendix A. Additionally, analytical sensitivity of ACTB was 0.1% and efficiency was 96% (Appendix A).

#### 4.2.5. Statistical Analysis

The Kolmogorov–Smirnov test was used to check for normality in distribution and the chi-squared test was used for comparison between discrete variables. One-way ANOVA test that was followed by Bonferroni post-hoc or Kruskal–Wallis test was applied to compare continuous variables between subgroups. In case of binary variables, t-test or Mann–Whitney test were also applied. Pearson or Spearman correlation was used for comparison between two continuous variables. Statistical significance was placed at *p*-value < 5 × 10^−2^. Statistical analysis was conducted with the IBM SPSS 19.0 statistical software (IBM Corp. 2010. IBM SPSS Statistics for Windows, Version 19.0. Armonk, NY, USA).

## 5. Conclusions

Presented data demonstrate *ENPP2* promoter hypermethylation in BrCa tissues associated with decreased expression, suggesting epigenetic regulation of its expression. Methylation events are correlated to BrCa progression and metastatic potential. In addition, we demonstrate that the *ENPP2* methylation assessed in liquid biopsy could offer a minimally invasive approach in early diagnosis and monitoring upon prospective evaluation. Our data introduce *ENPP2* methylation as a putative biomarker in BrCa.

## Figures and Tables

**Figure 1 ijms-23-03717-f001:**
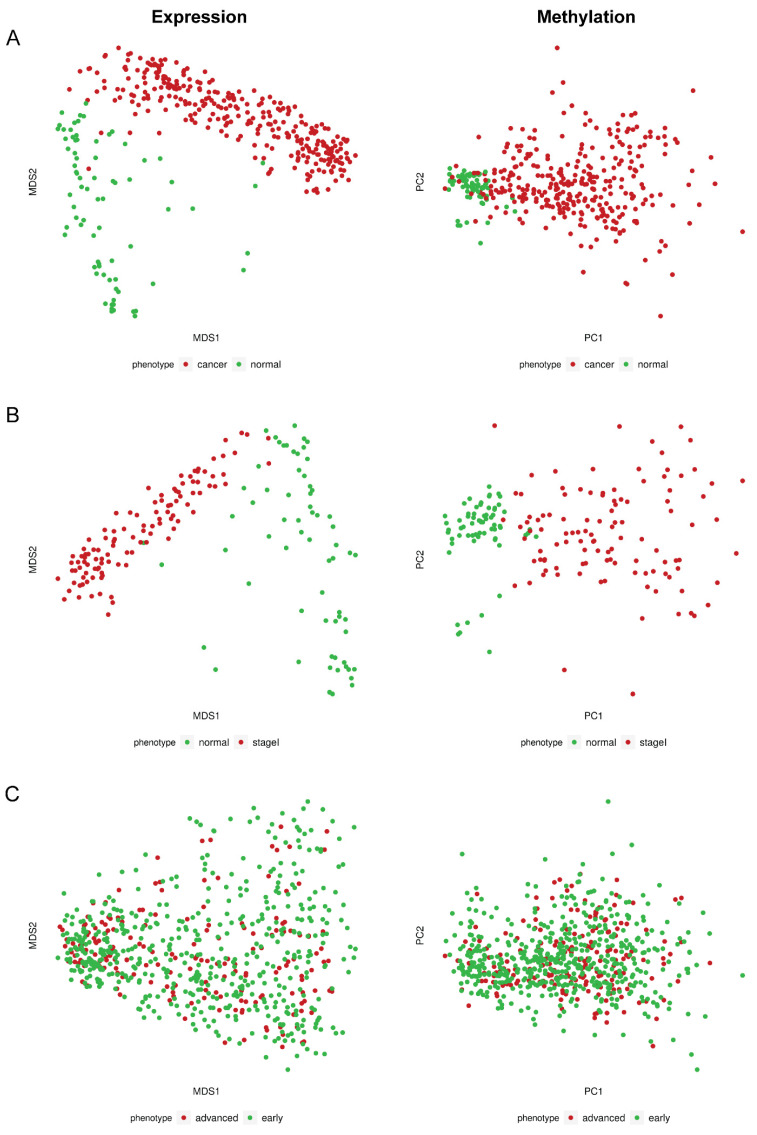
Dimensionality reduction plots for TCGA expression and methylation data. (**A**) MDS plot for normalized expression values of BrCa and normal breast tissue samples (**left**); PCA plot for level 3 beta methylation values of the same phenotypes (**right**). (**B**) MDS plot for normalized expression values of stage I BrCA and normal tissues (**left**); PCA plot for level 3 beta methylation values of the same phenotypes (**right**). (**C**) MDS plot for normalized expression values of advanced and early BrCA tissues (**left**); PCA plot for level 3 beta methylation values of the same phenotypes (**right**). MDS: Multidimensional scaling; PCA: principal component analysis.

**Figure 2 ijms-23-03717-f002:**
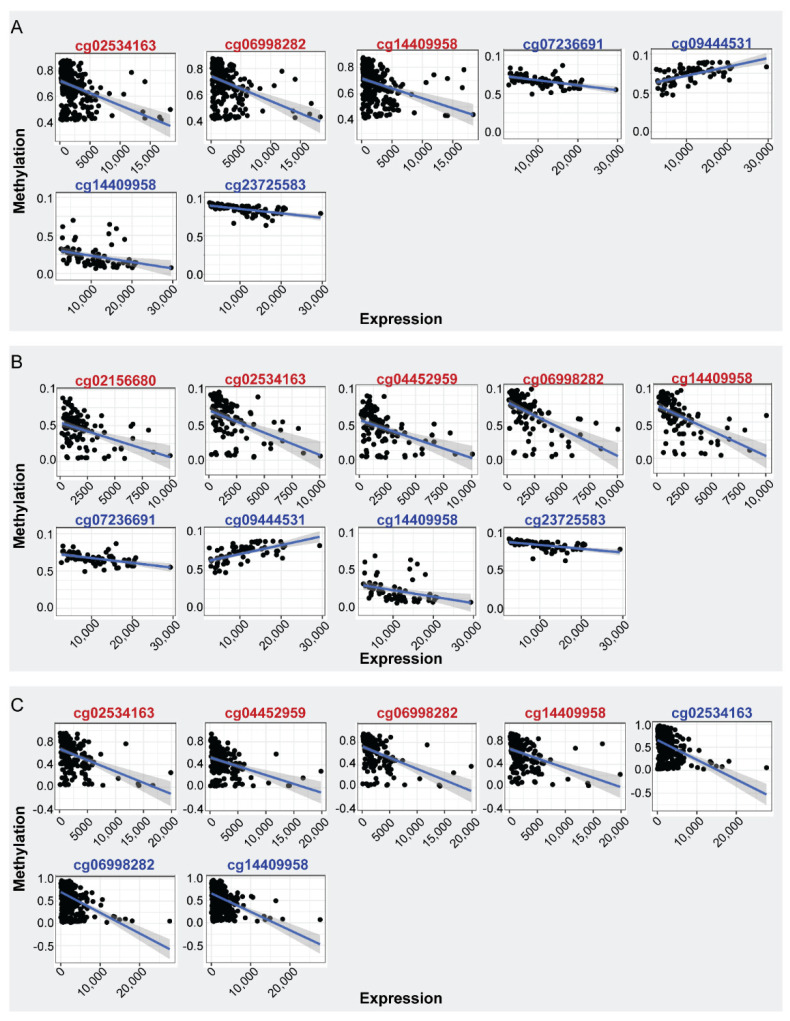
Spearman correlation of *ENPP2* CGs methylation and mRNA expression. CGs showing significant correlations are depicted (|rho| >= 0.40, FDR < 5 × 10^−2^). (**A**) Expression-methylation scatter plots of CG sites of BrCa (red) and normal (blue) samples, (**B**) expression-methylation scatter plots of CG sites of BrCa stage I (red) and normal (blue) samples, (**C**) expression-methylation scatter plots of CG sites of advanced (red) and early (blue) BrCa samples.

**Figure 3 ijms-23-03717-f003:**
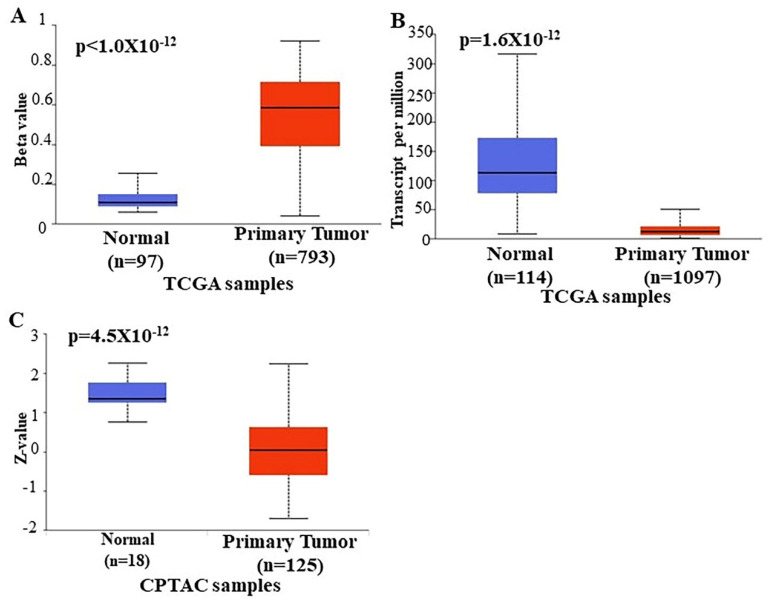
Analysis of *ENPP2* (**A**) promoter DNA methylation, (**B**) mRNA expression, and (**C**) ATX protein expression between primary BrCa tumors and normal tissues using the UALCAN platform. Abbreviations: TCGA: The Cancer Genome Atlas, CPTAC: Clinical Proteomic Tumor Analysis Consortium, BrCa: breast cancer.

**Figure 4 ijms-23-03717-f004:**
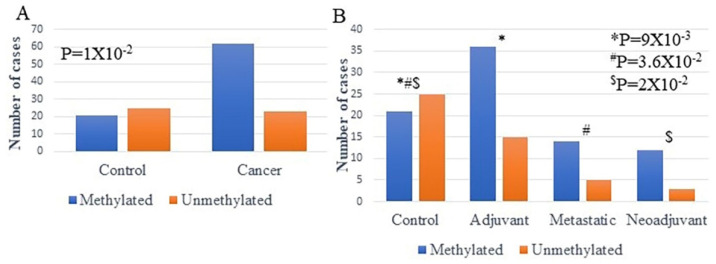
Methylation of *ENPP2* estimated by qMSP in ccfDNA (**A**) from BrCa patients and healthy individuals (**B**) from each group separately (adjuvant, metastatic, neoadjuvant, and healthy). Abbreviations: ccfDNA: circulating cell-free DNA; qMSP: quantitative methylation-specific PCR; CRC: colorectal cancer. * Control in relation to Adjuvant, # Control in relation to Metastatic, $ Control in relation to Neoadjuvant.

**Figure 5 ijms-23-03717-f005:**
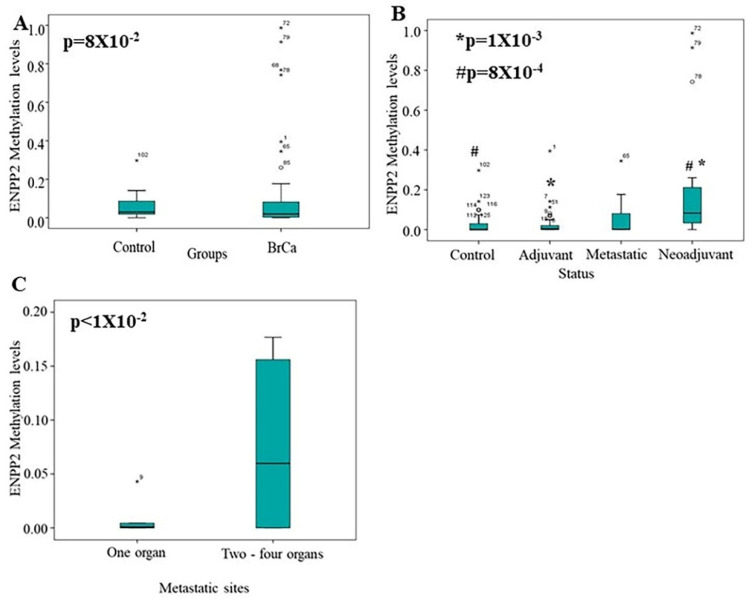
Median methylation levels of *ENPP2*. Boxplots depict methylation levels in ccfDNA of (**A**) BrCa compared to control group and (**B**) in each studied group separately; (**C**) BrCa patients with one metastatic site compared to those with two or more. Abbreviations: BrCa: breast cancer, * Adjuvant in relation to Neoadjuvant, # Control in relation to Neoadjuvant.

**Table 1 ijms-23-03717-t001:** *ENPP2* DMCs identified via in silico analysis of BrCa and normal breast tissues.

CG ID	Mβ Value Normal	Mβ Value BrCa	Δβ Value	Methylation in BrCa	Gene Region	FDR
Normal breast tissue vs. BrCa						
cg00320790	0.96	0.95	0.01	Down	Body	5.97 × 10^−4^
cg20048037	0.92	0.87	0.05	Down	Body	1.13 × 10^−12^
cg09444531	0.77	0.79	−0.02	Up	Body	5.16 × 10^−3^
cg26078665	0.77	0.84	−0.07	Up	Body	7.32 × 10^−14^
cg23725583	0.85	0.92	−0.06	Up	Body	1.03 × 10^−15^
cg02534163	0.06	0.53	−0.47	Up	1st Exon	3.15 × 10^−91^
cg04452959	0.03	0.44	−0.41	Up	TSS200	4.56 × 10^−80^
cg02709432	0.09	0.57	−0.48	Up	TSS200	6.71 × 10^−73^
cg02156680	0.04	0.44	−0.39	Up	TSS1500	9.18 × 10^−72^
cg06998282	0.09	0.62	−0.53	Up	TSS1500	9.54 × 10^−76^
Primary vs. Metastatic BrCa						
cg20048037	0.87	0.82	0.06	Down	Body	3.99 × 10^−2^
cg09444531	0.78	0.71	0.06	Down	Body	3.77 × 10^−2^
cg26078665	0.86	0.79	0.07	Down	Body	9.09 × 10^−4^
cg23725583	0.92	0.88	0.04	Down	Body	2.56 × 10^−2^
cg02534163	0.55	0.74	−0.19	Up	1st Exon	1.26 × 10^−4^
cg06998282	0.64	0.79	−0.15	Up	TSS1500	2.28 × 10^−3^
Normal breast vs. stage I BrCa						
cg20048037	0.92	0.89	0.03	Down	Body	4.35 × 10^−4^
cg09444531	0.77	0.80	−0.03	Up	Body	9.27 × 10^−3^
cg26078665	0.78	0.86	−0.08	Up	Body	1.07 × 10^−7^
cg23725583	0.86	0.93	−0.07	Up	Body	1.74 × 10^−8^
cg02534163	0.06	0.55	−0.48	Up	1st Exon	2.45 × 10^−49^
cg04452959	0.04	0.47	−0.43	Up	TSS200	5.53 × 10^−38^
cg02709432	0.10	0.61	−0.51	Up	TSS200	1.14 × 10^−38^
cg02156680	0.05	0.47	−0.43	Up	TSS1500	5.59 × 10^−39^
cg06998282	0.10	0.66	−0.56	Up	TSS1500	2.93 × 10^−35^
Early vs. Advanced BrCa						
cg01243251	0.94	0.92	0.014	Down	Body	3.10 × 10^−2^
cg20162626	0.75	0.69	0.051	Down	Body	3.12 × 10^−3^

Abbreviations: BrCa: breast cancer, DMCs: differentially methylated CpGs, FDR: false discovery rate, Mβ Value: mean β value, Δβ Value: difference between mean values.

**Table 2 ijms-23-03717-t002:** *ENPP2* differential expression analysis results based on TCGA data. |FC| > = 1.2 and FDR < 0.05 are considered as thresholds of significant deregulation.

Compared Breast Groups	Fold Change	*p*-Value	FDR
Cancer_vs_Normal	−5.15	1.18 × 10^−^^67^	3.96 × 10^−^^66^
StageI_vs_Normal	−5.46	6.28 × 10^−^^54^	3.43 × 10^−^^52^
Advanced_vs_Early	1.20	1.23 × 10^−2^	9.41 × 10^−2^

**Table 3 ijms-23-03717-t003:** Spearman correlation coefficient between *ENPP2* CG methylation and mRNA expression for each of our clinical endpoints. CGs showing significant correlations are depicted (|rho| >= 0.40, FDR < 5 × 10^−2^).

BrCa vs. Normal					
Tissue	CG	Gene Region	Rho	FDR	Correlation
BrCa	cg02534163	First Exon	−0.40	1.18 × 10^−^^12^	Negative
	cg06998282	TSS1500	−0.42	1.53 × 10^−^^13^	Negative
	cg14409958	TSS1500	−0.42	2.10 × 10^−^^13^	Negative
Normal	cg09444531	Body	0.62	4.77 × 10^−^^06^	Positive
	cg23725583	Body	−0.70	3.02 × 10^−^^11^	Negative
	cg07236691	Body	−0.55	1.27 × 10^−^^08^	Negative
	cg14409958	TSS1500	−0.52	9.53 × 10^−^^07^	Negative
Stage I BrCa vs. Normal					
Stage I	cg02534163	First Exon	−0.46	1.54 × 10^−06^	Negative
	cg04452959	TSS200	−0.44	3.74 × 10^−06^	Negative
	cg14409958	TSS1500	−0.64	3.29 × 10^−13^	Negative
	cg02156680	TSS1500	−0.42	1.28 × 10^−05^	Negative
	cg06998282	TSS1500	−0.63	6.04 × 10^−13^	Negative
Normal	cg09444531	Body	0.64	3.25 × 10^−08^	Positive
	cg23725583	Body	−0.66	1.36 × 10^−08^	Negative
	cg07236691	Body	−0.53	1.40 × 10^−05^	Negative
	cg14409958	TSS1500	−0.51	4.25 × 10^−05^	Negative
Early vs. Advanced BrCa					
Early	cg02534163	First Exon	−0.42	2.56 × 10^−23^	Negative
	cg06998282	TSS1500	−0.47	4.588 × 10^−29^	Negative
	cg14409958	TSS1500	−0.46	5.37 × 10^−28^	Negative
Advanced	cg02534163	First Exon	−0.43	1.23 × 10^−09^	Negative
	cg04452959	TSS200	−0.41	5.47 × 10^−^^09^	Negative
	cg14409958	TSS1500	−0.48	4.72 × 10^−^^12^	Negative
	cg06998282	TSS1500	−0.49	4.18 × 10^−^^12^	Negative

Abbreviations: FDR: false discovery rate, TSS: Transcription Start Site.

**Table 4 ijms-23-03717-t004:** *ENPP2* DMCs identified between ductal and lobular BrCa via in silico analysis of TCGA cases.

CG ID	Mβ Value Ductal Cancer	Mβ Value Lobular Cancer	Δβ Value	Methylation in Ductal Cancer	Gene Region	FDR
cg01243251	0.94	0.93	0.01	Up	Body	1.53 × 10^−2^
cg20048037	0.86	0.90	−0.04	Down	Body	1.11 × 10^−2^
cg02156680	0.47	0.52	−0.04	Down	TSS1500	2.52 × 10^−2^

Abbreviations: BrCa: breast cancer, DMCs: differentially methylated CpGs, FDR: false discovery rate, Mβ Value: mean β value, Δβ Value: difference between mean values.

**Table 5 ijms-23-03717-t005:** *ENPP2* DMCs identified in ccfDNA of BrCa patients and healthy individuals via in silico analysis.

CG ID	Mβ Value BrCa	Mβ Value Normal	ΔβValue	Methylation in BrCa	Location	FDR
cg07236691	0.583	0.817	−0.234	Down	Body	3.04 × 10^−3^
cg20048037	0.711	0.331	0.380	Up	Body	5.08 × 10^−5^
cg20162626	0.489	0.814	−0.325	Down	Body	3.18 × 10^−4^
cg02534163	0.802	0.206	0.596	Up	1st Exon	6.30 × 10^−11^
cg04452959	0.620	0.020	0.599	Up	TS200	2.51 × 10^−13^
cg02156680	0.515	0.028	0.487	Up	TSS1500	1.48 × 10^−13^
cg06998282	0.772	0.057	0.715	Up	TSS1500	1.61 × 10^−8^
cg14409958	0.748	0.053	0.695	Up	TSS1500	2.24 × 10^−6^

Abbreviations: BrCa: breast cancer, DMCs: differentially methylated CpGs, FDR: false discovery rate, Mβ Value: mean β value, Δβ Value: difference between mean values.

**Table 6 ijms-23-03717-t006:** Comparisons, endpoints, study group characteristics, and clinical significance of the tissue datasets used in the bioinformatic analysis. Abbreviations: BrCa = breast cancer, NR = Not Relevant.

Study Groups	Tissues	Age (Years) Median (Range)	Stage	Significance
1. BrCa vs. Normal	520 BrCa (primary and metastatic)	49 (26–80)	102 Stage I 264 Stage II 114 Stage III 40 Stage IV	Diagnosis
	185 Normal	47 (26–80)	NR	
2. Primary vs. Metastatic BrCa	132 PrimaryBrCa	55 (47–55)	22 Stage I 75 Stage II 35 Stage III	Diagnosis/Prognosis
	31 Metastatic BrCa	54 (41–80)	31 Stage IV	
3. Stage I BrCa vs. Normal	136 Stage I BrCa	54 (27–80)	136 Stage I	Diagnosis/Prognosis
	111 Normal	58 (29–80)	NR	
4. Early vs. Advanced BrCa	521 EarlyBrCa	58 (26–80)	115 Stage I 406 Stage II	Diagnosis/Prognosis
	221 Advanced BrCa	55 (27–80)	221 Stage III	

**Table 7 ijms-23-03717-t007:** Demographic and clinicopathological characteristics of BrCa and control groups.

Group	Total	Adjuvant	Metastatic	Neoadjuvant	Control
N	132	52	19	15	46
Age					
Mean (±SD)	57.7 (±13.9)	58.7 (±12.0)	61.9 (±9.8)	55.5 (±16.6)	55.6 (±13.7)
Median (range)	59.0 (0.0–83.0)	60.5 (27.0–80.0)	65.0 (44.0–75.0)	51.0 (29.0–79.0)	57.0 (26.0–83.0)
Grade					
1	10	10	-	-	
2	25	19	-	6	
3	30	16	8	6	
N/A	21	7	11	3	
Stage					
I	15	15	-	-	
II	28	28	-	-	
III	9	9	-	-	
IV	19	-	19	-	
N/A	15	-	-	15	
Lymphnode status					
Negative	27	24	-	3	
Positive	33	26	-	7	
N/A	26	2	19	5	
Tumor size (before surgery)					
≤2	30	25	-	5	
>2 and ≤6.5	33	26	-	7	
N/A	23	1	19	3	
Metastatic sites					
Lung	12	-	12	-	
Skin	1	-	1	-	
Distantlymphnodes	5	-	5	-	
Pancreas	1	-	1	-	
Bone	9	-	9	-	
Liver	4	-	4	-	
Pleural	1	-	1	-	

Abbreviations: BrCa = breast cancer, SD = standard deviation, N/A = Non-Available or Non-Applicable.

**Table 8 ijms-23-03717-t008:** Primer sequences, annealing temperatures, and genomic locations used for qMSP assays.

GENE	Primer Sequence (5′–3′)	Annealing Temperature (°C)	Product Length	Genomic Loci
*ENPP2*	MET F: CGTTTTTTTATTTGATACGATTGGAACGA MET R: CAAAACCTCAAAACAATACACTCCGTAA	60	117bp	Chr8: 120650976- 120651092 (+1 strand)
*ACTB*	F: TGGTGATGGAGGAGGTTTAGTAAG R: AACCAATAAAACCTACTCCTCCC	60	134bp	chr7: 5558705– 5558838 (−1 strand)

Abbreviations: MET: methylated, F: forward, R: reverse.

## Data Availability

Data are available upon request.

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
