# Peer review of "ENPP2 Promoter Methylation Correlates with Decreased Gene Expression in Breast Cancer: Implementation as a Liquid Biopsy Biomarker"

_ijms, 2022, doi:10.3390/ijms23073717_

Round 1
Reviewer 1 Report
.

Author Response
"Please see the attachment."

Reviewer 2 Report
Dear Dr. Chatzaki,
The article under review has some serious editing errors to be remove before resubmission.
- Formatting errors are very frequently identified, Few examples, BUT NOT LIMITED are as follows:
a. line 79 450kplatformcontainsby….is condensed in text…should be re write separately
b. Line 98 BrCatissues should be corrected
c. Line 102 s betweenENPP2
d. Line 128 ifaberrant
e. Line 139 correlationbetween and expressionwas
f. Iine 146 Table 1. ENPP2 DMCsidentified via in silicoanalysis ofBrCa and normal breast tissues.
2. Graphs has to be improved with correct format and please check the format and fonts size and style presented into each graph..e.g. in one graph Figure 5 A and B the X is shown in three different formats.
3. All the Table captions having frequent error of text condensation
4. There are no error bar and Axis title presented in Figure 4
5. References are not written according to journal format.
These are few examples of the editing mistakes, I would suggest to resubmit after extensive editing and formatting this article.
Best regards,
Saima
Author Response
"Please see the attachment."

Round 2
Reviewer 1 Report
The authors responded satisfactorily to the concerns related to the typification of breast cancers. The authors point out that some existing databases do not contain relevant information (as it occurs in this case where patients are not grouped with enough clinical and pathological information). In my opinion, the explanations and corrections are sound and improve the manuscript's quality and interest.
Thank you for sufficiently enhancing the quality of the figures and for taking care of the minor corrections suggested.
Overall, the manuscript is now more sound, readable and exciting.